# Harnessing the Role of Bacterial Plasma Membrane Modifications for the Development of Sustainable Membranotropic Phytotherapeutics

**DOI:** 10.3390/membranes12100914

**Published:** 2022-09-22

**Authors:** Gayatree Panda, Sabyasachi Dash, Santosh Kumar Sahu

**Affiliations:** 1Department of Biotechnology, Maharaja Sriram Chandra Bhanjadeo University (Erstwhile: North Orissa University), Baripada 757003, India; 2Department of Pathology and Laboratory Medicine, Weill Cornell Medicine, New York, NY 10065, USA

**Keywords:** multi-drug resistance, lipopolysaccharides, lipid A, aminoacylation, membranotropic phytochemicals

## Abstract

Membrane-targeted molecules such as cationic antimicrobial peptides (CAMPs) are amongst the most advanced group of antibiotics used against drug-resistant bacteria due to their conserved and accessible targets. However, multi-drug-resistant bacteria alter their plasma membrane (PM) lipids, such as lipopolysaccharides (LPS) and phospholipids (PLs), to evade membrane-targeted antibiotics. Investigations reveal that in addition to LPS, the varying composition and spatiotemporal organization of PLs in the bacterial PM are currently being explored as novel drug targets. Additionally, PM proteins such as Mla complex, MPRF, Lpts, lipid II flippase, PL synthases, and PL flippases that maintain PM integrity are the most sought-after targets for development of new-generation drugs. However, most of their structural details and mechanism of action remains elusive. Exploration of the role of bacterial membrane lipidome and proteome in addition to their organization is the key to developing novel membrane-targeted antibiotics. In addition, membranotropic phytochemicals and their synthetic derivatives have gained attractiveness as popular herbal alternatives against bacterial multi-drug resistance. This review provides the current understanding on the role of bacterial PM components on multidrug resistance and their targeting with membranotropic phytochemicals.

## 1. Introduction

The worrisome increase in bacterial resistance to current generation of most efficacious antibiotics is a rapidly expanding global health problem. Currently, at least 700,000 people die each year due to antimicrobial drug-resistant pathogens, which is expected to rise exponentially to 10 million by 2050, which could displace more than 20 million people below the poverty line by 2030 [1,2,3]. The present United Nation and World Health Organization reports also cite the dire need for innovative and sustainable methods for the development of novel drugs to overcome antimicrobial resistance diseases [4,5]. Antimicrobials either target the cell envelope or cytosolic components of bacteria to inhibit essential cellular events. While lipophilic antibiotics (e.g., macrolides) can diffuse directly through the bacterial plasma membrane (PM), low molecular weight hydrophilic antibiotics (e.g., β-lactams) having size exclusion limit < 600 Da use porins to reach their intracellular targets [6]. The cell envelope in Gram-negative bacteria is made of an outer membrane (OM), peptidoglycan cell wall (CW), and inner membrane (IM) (Figure 1). The OM is heterogeneous and asymmetric with an outer leaflet consisting of lipopolysaccharide (LPS) (or lipooligosaccharide) (LOS) and an inner leaflet consisting of phospholipids (PLs). However, the IM is a symmetric phospholipid (PL) bilayer. The cell envelope of Gram-positive bacteria such as *S*. *aureus* has a single plasma membrane made of PLs followed by a thick cell wall with no periplasmic space. Efficient permeation of antimicrobials through cell envelope is rate limiting in their activity [7]. As the cell wall is porous and permeable to most antibiotics, permeability through the OM and IM are rate limiting factors in determining their cellular bioavailability in Gram-negative bacteria. However, in Gram-positive bacteria such as *S*. *aureus*, permeability through plasma membrane is crucial for antimicrobial efficacy [8].

Antimicrobials that target plasma membrane are amongst the most effective class of antibiotics against drug-resistant bacteria because: (i) Most conventional antimicrobials target cytosolic pathways such as cell division or biosynthetic machineries. However, drug resistant bacteria are either quiescent or slow-growing in nature, making their elimination cumbersome due to lack of abundant target sites [9]. (ii) Normally, drugs with cytosolic targets are polar with poor membrane permeability, resulting in significant of their effective cytosolic concentration [10]. (iii) Bacterial plasma membrane harbors a large repertoire of conserved lipids and proteins; those can be targeted without developing multi-drug resistance [11].

Cationic antimicrobial peptides (CAMPs) are secreted by the animal immune system as defense molecules against infectious bacterial pathogens. These are 10–50 amino acids (aa) in length amphiphilic peptides with +2 to +7 unit positive charges and ~50% non-polar aa that exhibit membranolytic activity against broad spectrum of bacteria [12]. Anionic lipopeptides (e.g., daptomycin) and glycopeptides (e.g., vancomycin) are also efficient disruptors of bacterial plasma membrane, however, with yet unclear mechanisms of action [13]. All these peptides, irrespective of their charge, exhibit high binding affinity to the negatively charged bacterial plasma membrane leading to their disruption. However, repeated exposure to CAMPs leads to modification of plasma membrane lipids, resulting in CAMP-resistant bacteria [14]. These membrane modifications include (i) increased charge, (ii) altered fluidity, (iii) reduction in target lipids, (iv) modification of target lipids, or (v) altered membrane domain organization. However, many drug-resistant bacteria use multiple membrane-mediated mechanisms to evade the most potent antimicrobial drugs [15].

A wide range of natural and synthetic amphiphilic compounds that disrupt bacterial plasma membrane are promising herbal alternatives to multi-drug-resistant bacteria [16,17,18,19]. However, many of these compounds exhibit significant hemolytic activity at their minimal inhibitory concentration (MIC) [20]. Hence, design of novel membrane-targeting molecules that exhibit high specificity towards bacterial plasma membrane are either potent antibacterial drugs or adjuvant therapeutics that increases drug permeability into bacteria [21]. Phytochemicals such as glycosides and saponins with high membrane-penetrating ability are being explored as eco-friendly herbal alternatives of synthetic antimicrobials [22]. Membranotropic phytochemicals exhibit their antibacterial action through (i) alteration of membrane fluidity, (ii) increase in membrane permeability, (iii) formation of membrane microdomains, (iv) induction of membrane lipid flip-flop [23], (v) binding to lipid components [16], or (iv) affecting membrane protein function [24]. However, a membranotropic phytochemical might affect multiple of the above membrane-associated activities. The mechanistic details of many antibacterial phytochemicals remain elusive. It is not yet clear if there exists a structure–function correlation between different phytochemical groups and their membranotropic activity.

In this review, we provide a comprehensive analysis of recent developments on bacterial membrane lipidome modifications leading to resistance against membrane-targeted antimicrobials. Further, we discuss how membranotropic phytochemicals could be explored as potent lead compounds for development of novel antibacterial therapeutics to fight MDR bacteria.

## 2. Role of OM in Gram-Negative MDR Bacteria

Four of the “ESKAPE” group of drug-resistant human pathogens, e.g., *Enterococcus faecium*, *Staphylococcus aureus*, *Klebsiella pneumoniae*, *Acinetobacter baumannii*, *Pseudomonas aeruginosa*, and *Enterobacter* sp. are Gram-negative. The OM of these bacteria is a sophisticated macromolecular assembly of LPSs, PLs, porins, and other OM-associated proteins. LPS, a peculiar glycolipid unique to Gram-negative bacteria is the hallmark of OM that consists of a glucosamine-based lipid (lipid A) attached to a core oligosaccharide followed by a distal polysaccharide chain (Figure 2). The OM is asymmetric with LPS residing exclusively to the exoplasmic (outer) leaflet and PLs residing to the cytosolic (inner) leaflet. The strain-specific immune-reactivity and virulence of most Gram-negative bacteria is because of the immensely variable composition and organization of the sugar moieties and fatty acyl tails of LPS molecules. This incredible variability in LPS that is attained through unique organization and adaptability to multiple secondary covalent modifications plays a key role in the MDR of Gram-negative bacteria.

### 2.1. Asymmetric Organization of LPS in Gram-Negative OM LEADS to CAMP Resistance 

The asymmetric distribution of LPS in the outer leaflet of Gram-negative OM provides an intrinsic resistance against most bulky hydrophobic and cationic antimicrobials. LPS has a nearly conical geometry with its membrane-embedded hexa-acylated lipid A and a highly variable bulky polysaccharide chain that extends ~30 A° beyond the plane of the OM (Figure 2). The LPS layer is a robust and continually evolving antibiotic barrier leading to bacterial MDR and virulence [25]. An in vitro reconstituted model of the Gram-negative OM suggests that the trans-bilayer asymmetry is crucial for stable interaction and insertion of cationic peptides [26].

The asymmetry of Gram-negative OM is acutely regulated by a set of three different proteins that work in concert, namely, the OM phospholipase A_2_ (PldA), the LPS, palmitoyltransferase (PagP), and the maintenance of outer membrane lipid asymmetry (Mla) system [27]. Both PldA and PagP are involved in degradation or removal of mislocalized PLs in the outer leaflet of OM. Mla is a six-component lipid flippase complex conserved in all Gram-negative bacteria that facilitates the ATP-dependent retrograde transport of mislocalized PLs from the outer leaflet of OM to the IM (Figure 1). Mla complex contributes to the the broad-spectrum antibiotic resistance in several pathogens, most prominently in *A*. *baumannii* [28]. MlaA, the subunit that is localized to the inner leaflet of OM, has the shape of a doughnut that only allows the removal of mislocalized lipids in the outer OM leaflet without affecting the inner leaflet lipids [29]. Structural elucidation of the Mla complex reveals that PLs from MlaA are translocated to MlaC that shuttles through the peptidoglycan layer and periplasm to deliver the PLs to the IM-localized MlaBDEF complex. PLs from the binding site of MlaC are transferred to the bowl-shaped domain formed by the hexameric D subunits of the MlaBDEF complex on its periplasmic site. PLs are then translocated to the inner-IM leaflet through a gated transmembrane tunnel formed by the helices of D and E subunits. The hydrophobic side chains of Leu^153^/Leu^154^ directed towards the lumen of the tunnel plausibly facilitates the movement of the non-polar fatty acyl tails of PLs. However, movement of the polar head group is facilitated by the polar cationic side chains of R14, R47, and R234 located toward the cytosolic end of the tunnel (Figure 1) [30]. PLs from the bifurcated cytosolic outlet of MlaBDEF complex diffuse into the inner-IM leaflet. PLs localized to the inner-IM leaflet is then equilibrated with the outer leaflet by ATP-independent biogenic membrane flippases [31]. However, it remains to be explained how the PLs is transferred between MlaA, MlaC and MlaD. Although the Mla complex was initially proposed to be involved in the retrograde transport of PLs from the outer OM leaflet to the inner IM leaflet, it is becoming increasingly clear that the same complex also facilitates the anterograde translocation of PLs from the IM to the OM in LPS-deficient strains of *A*. *baumanii* [32]. Mla in OM of Gram-negative bacteria is essential for the maintenance of the barrier function of OM to protect the bacterium againstthe noxious compounds. Hence, Mla inhibitors could be explored as a potent antibacterial that may act alone or in combination [33,34]. 

### 2.2. Loss of OM Asymmetry in Gram-Negative Bacteria

The negatively charged LPS of Gram-negative OM is the substrate for attachment of CAMPs such as polymyxins to bacteria. The cationic antibiotic polymyxin E (colistin) inhibits divalent cation-mediated bridging between adjacent LPS molecules, leading to their loss in the outer-OM leaflet [35]. In *A. baumanii*, a mutant non-functional *lpxA* is responsible for the loss of LPS which otherwise acts as the target for colistin. It also exhibits increased sensitivity towards other clinically relevant antibiotics such as Cefepime, Teichoplanin, and Azithromycin [36]. However, repeated exposure of *A. baumanii* to colistin stimulates loss of LPSs in outer-OM leaflet, leading to their replacement with PLs resulting in loss of OM asymmetry [28]. In wild-type *A. baumannii*, combined action of the periplasmic lipid-shuttle complex Mla and phospholipase A (PldA) works to remove mislocalized PLs in OM and maintains its asymmetry. However, colistin resistant variants of *A. baumannii* are nonfunctional mutants of *Mla* and *PldA* that compensate for the loss of LPS in OM outer leaflet by stably replacing them with PLs [25]. These LPS-deficient phenotypes of *A. baumannii* demonstrate enhanced resistance to colistin (Appendix A).

However, interaction of polymyxins with PLs in the inner IM leaflet is not yet completely understood. Recent investigation reveals that MCR-1-mediated colistin resistance in *Escherichia coli* is due to modified LPSs at the cytoplasmic membrane (CM) rather than OM [37]. LPS that is synthesized in the inner IM leaflet is flipped to the outer leaflet from which it is transported to the OM outer leaflet by a LPS transport complex known as LptABCDEFG (Li et al., 2020). Hence, murepavadin, an inhibitor of LPS transport machinery that leads to accumulation of LPS in CM, kills *E. coli* by lysing CM [38].

### 2.3. Covalent Modification of LPS in MDR

Bulky CAMPs such as polymyxin B (MW = 1.2 kDa) with size > 600 Da target bacterial OM, leading to its disruption resulting in cell lysis [39,40]. Some Gram-negative bacteria like *E. coli* can acquire antibiotic resistance by modifying the sugar content in the OM [41]. OM of many Gram-negative bacteria such as *E*. *coli*, *S*. *enterica*, *S*. *typhimurium,* and *K*. *pneumoniae* modify their LPS with additional functional groups such as fatty acyl chains (e.g., palmitate and S-2-hydroxymyristate), phosphoethanolamine (pEtN), 4-amino-4-deoxy-L-arabinose (L-Ara4N), and glycosyl groups to enhance their resistance against membrane-targeted CAMPs such as polymyxins [42] (Figure 2). Although LPS can have varying number of sugar units, studies indicate that the core oligosaccharide portion, especially the first heptose in the LPS, is critical for conferring antibiotic resistance in gram-negative bacteria [43].

Enzymes that modify lipid A are regulated by two conserved component systems (TCS), PmrAB and PhoPQ, in response to specific environmental signals. PmrAB is activated in response to CAMPs, high Fe^3+^ and acidic pH, whereas PhoPQ is activated in response to divalent cations (e.g., Ca^2+^/Mg^2+^) or CAMPs. PmrAB activates eptA (also known as pmrC) and arabinose (*arn*) operon that encode pEtN and L-Ara4N transferases, respectively. PhoPQ phosphorylates the transcription factors resulting in transcription of PagL (only in *Salmonella*) and PagP that, respectively, remove or add fatty acyl groups to lipid A. In addition, PhoPQ directly activates *arn* expression in *Klebsiella* and *Yersinia* sp. [44]. Both PhoPQ and PmrAB are interdependent and are connected through PmrD that binds to phospho-PmrA to prevent B-mediated dephosphorylation of transcription factors. Constitutive expression of PmrA dependent genes that increases modification of lipid A with pEtN and l-Ara4N, resulting in MDR of Gram-negative bacteria.

#### 2.3.1. Fatty Acylation of Lipid A

Lipid A, the OM-embedded portion of LPS, also termed as endotoxin, is responsible for most of the pathophysiological effects associated with Gram-negative sepsis. Lipid A potently activates the host innate immune system leading to secretion of CAMPs, cytokines, clotting factors, and immunostimulatory molecules. It is a ubiquitous component of Gram-negative OM, making its modification a conserved resistance mechanism against most membrane-targeted antimicrobials. The length, number, and distribution of acyl chains are the key factors contributing to the biological activity of lipid A [45]. The hexa-acylated lipid A can be modified by addition of a palmitate in acyloxyacyl linkage at position 2 of hexosamine sugars to produce hepta-acylated lipid A (Figure 2). The hepta-acylated lipid A enhances the resistance of *A*. *baumannii*, *E. coli* and *Salmonella* to the last resort of CAMPs (e.g., colistin) by reinforcing the cell surface LPS [46]. Enhanced acylation of lipid A (PagP-dependent or independent) is a general process reported in *E. coli*, *Y. enterocolitica*, *B. pertussis,* and *A. baumannii*. PagP cleaves the mislocalized PL in OM, restoring the outer leaflet composition, thereby increasing the bacterial resistance to aminoglycosides by 4-fold [47]. Secondary acylation of lipid A with laureate and myristate in *K. pneumoniae* increases its resistance against polymixin B and colistin. It also increases the virulence limit of bacterial pathogens against host immune response and survival on the desiccated surfaces of medical appliances [48].

Hepta-acylation of lipid A in *E. coli* and *Salmonella* is regulated by the gene PagP. However, *A*. *baumannii* uses the two-component system LpxL and LpxM for lipid A acylation. In *E. coli*, PagP-dependent lipid A palmitoylation increases the hydrophobic van der Waals forces of the LPS layer that prevent CAMP insertion. Hepta-acylation of lipid A protects *Salmonella* from vertebrate CAMPs such as C18G, an α-helical membranolytic peptide [49]. In *E*. *coli* and *S*. *enterica*, PagP transfers a palmitate from phosphatidylethanolamine to the R-2-hydroxymyristate of lipid A, leading to generation of hepta-acylated lipid A. However, the position of the palmitoylation may vary in different bacteria. PagP palmitoylates the *N*-linked *R*-3-hydroxymyristate chain at position 2 of lipid A in Enterobacteria [50]. In *Bordetella,* PagP palmitoylates the O-linked chains at position C–3′, C–2 and C–3 of the hexosamine residue in lipid A, which increases membrane rigidity. *Pseudomonas aeruginosa* palmitoylates lipid A at position 3′ [51,52]. A *pagP* homolog from *Legionella pneumophila* was identified as *rcp* (resistance to cationic antimicrobial peptides) [53]. Host immune suppression is a key virulence strategy used by various drug-resistant bacteria. In *Salmonella enterica*, lipid A palmitoylation impairs the production of CAMPs through the suppression of TLR4 pathway [54]. In *Salmonella*, myristoylation of lipid A increases its resistance against polymyxin. This secondary acylation of lipid A is important for the addition of L-Ara4N to its phosphate groups, leading to a twofold resistance against polymyxins [55]. Asymmetric Langmuir model bilayers of outer leaflets and inner leaflets, consisting of LPSs and PLs, respectively, demonstrate that packing of the hydrophobic fatty acyl tails in LPS is the primary determinant of polymyxin B-induced OM disruption [56]. The OM is made rigid by divalent cation interactions, which raises the transition temperature (Tm) of the membrane [56].

As amide linkages are more rigid, planar, and stereochemically constrained, an increased number of amide-linked acyl chains in lipid A enhances membrane rigidity. For instance, the addition of three amide-linked acyl chains and only one ester-linked acyl chain instead of two ester- and two amide-linked acyl chains to lipid A in *C. jejuni* increases membrane rigidity. Accordingly, additional amide linkage influences the biological activity of lipid A by altering TLR4 response and enhanced CAMP resistance [57]. The palmitoylation of lipid A allows for increased hydrophobic interactions between neighboring LPS molecules, leading to increased resistance to membranolytic antimicrobials [58].

#### 2.3.2. Amino Glycosylation

Decoration of lipid A with cationic or zwitterionic groups such as L-4-aminoarabinose and phosphoethanolamine (PE) that shield the negative charge on phosphates leads to drug resistance in Gram-negative bacteria. First, it reduces binding of CAMPs to the outer leaflet of OM by reducing the anionic charge on the OM lipids. Second, it reduces the requirement of divalent cations (e.g., Ca^2+^ and Mg^2+^) that is essential for the intermolecular bridging between LPSs, leading to its stabilization [59]. In members of enterobacteriaceae such as *E*. *coli* and *Salmonella*, the PmrAB two-component system directly activates L-Ara4N biosynthesis, leading to addition of the cationic amine moiety to lipid A. However, some members of enterobacteriaceae family such as *Enterobacter cloacae* use the PhoPQ two-component system that contributes to colistin heteroresistance. PhoPQ directly binds the ArnB promoter to activate the L-Ara4N biosynthesis, its covalent attachment to Lipid A in IM, and transport to the outer leaflet of OM [44]. Although several factors, such as low Mg^2+^ concentration, low pH, presence of antibiotics, and high temperature are proven to modify lipid A, de novo expression of colistin resistance conferring peptides (Dcr) in *E*. *coli* have been proven to activate the *PmrAB* conferring colistin resistance [60]. In *E*. *coli* and *K*. *pneumoniae* strains, the mobile colistin-resistant gene 1 (mcr1) regulates collistin resistance to overcome destabilization of the bacterial outer membrane and prevents cell lysis [61]. However, absence of *mcr1* in colistin and carbapenem resistant strains of *K*. *pneumonia* with mutations in several *Arn* genes (e.g., *ArnA_DH/FT, UgdH, ArnC* and *ArnT*) reveals altered L-Ara4N biosynthetic pathways. The L-Ara4N precursor, undecaprenyl phosphate-α-L-Ara4-formyl-N is synthesized on the cytosolic side of IM from the undecaprenyl phosphate and UDP-L-Ara4-Formyl-N that is flipped to the periplasmic side. However, the flippase protein that facilitates this essential biosynthetic step remains elusive. Once on the periplasmic side of IM, the undecaprenyl phosphate-α-L-Ara4-formyl-N is deformylated and the L-Ara4-N moiety is transferred to lipid A of LPS [62]. Transfer of L-Ara4N to lipid A to the inner IM leaflet is reversibly catalyzed by the IM-associated integral protein ArnT [63]. ArnT exhibit a conserved topology in both *Burkholderia cenocepacia* and *Salmonella enterica* with its N-terminal domain forming 13 transmembrane helices and a globular periplasmic C-terminal domain [64]. From the IM, L-Ara4N-LPS is vectorially translocated to the OM through the periplasm that is catalyzed by the ATP-dependent ABC-transporter. This translocation requires an inter-membrane protein bridge, which connects both IM and OM that facilitates the unidirectional transport of L-Ara4N-LPS to the OM [65]. L-Ara4N is cationic and results in lowering of the negative charge on LPS leading to its reduced binding affinity to CAMPs. As the LPS-modification enhances pathogenicity and innate immunity evasion, ArnT that catalyzes transfer of L-Ara4N to lipid A can be a virulence factor which is an ideal target for development of membrane-targeted therapeutics. Overexpression of *pmrA* (polymyxin resistance gene *A*) that modifies lipid A with L-Ara4N results in 75% lowering of polymixin binding [66]. Similarly, deacetylation, another key modification of lipid A, is facilitated by several membrane-associated enzymes such as PagL, LpxR, NaxD, and YdjC. The glycylation, another lipid A modification that adds glycine or diglycine residues is reported to enhance drug resistance in Gram-negative bacteria. Phosphorylation or phosphate modification leads to secondary alteration of the core oligosaccharide structure and acylation status of lipid A, resulting in bacterial resistance against CAMPs like Polymixin B and Colistin [67].

## 3. Role of Phospholipid Modification in Bacterial MDR

### 3.1. PL Modification in OM

Gene PhoPQ was proven to increase levels of palmitoylated acyl-PG and CL, both of which are glycerophospholipids found in the inner leaflet of the Gram-negative OM. Combined regulation of glycerphospholipids and lipid A modification mediates CAMP resistance through alteration of surface charge and hydrophobicity of OM. These modifications either decrease binding of CAMPs to the OM or slow down their permeability through OM [68]. PhoPQ induces the transmembrane protein PbgA to transfer CL from the IM to the OM of Gram-negative bacteria. The globular region of PbgA binds to CL near the inner membrane and mediates CL transport to the outer membrane. Mutants that lack this globular region are less virulent [55].

### 3.2. PL Modification in Inner (Cytoplasmic) Membrane

In Gram-negative bacteria, PLs are the building blocks of the IM and inner OM leaflet. However, in Gram-positive bacteria, PLs form the only membrane. In most bacteria, major PLs are Phosphatidylglycerol (PG), Phosphatidylethanolamine (PE), and Cardiolipin (CL). In *E*. *coli*, IM has ~80% phosphatidyl ethanolamine (PE), 15% phosphatidyl glycerol (PG), and 5% cardiolipin (CL). In *S*. *aureus*, the PM is made from 60% PG and 40% CL, resulting in a negatively charged plasma membrane [69]. The negatively charged PM in many bacteria attracts positively charged CAMPs that result in membrane destabilization and subsequent lysis. However, many bacteria such as *S. aureus* decorate their anionic lipids, such as PG and CL with aminoacyl residues (e.g., lysine, alanine, arginine, and ornithine) to mask negative charge on their PM. The reduced negative charge on their PM enables them to escape the binding of CAMPs. Additionally, CLs and other membrane components form clusters termed as microdomains to encounter stress conditions such as high osmolarity, extreme temperature, and presence of antibiotics. Bacteria such as *S. aureus* contain specialized carotenoid pigments termed as staphyloxanthin that interact with membrane components (e.g., flotillins) to form microdomains.

In methicillin-resistant *Staphylococcus aureus* (MRSA), aminoacylation of PG and CL is facilitated by a multiple peptide resistance factor (MPRF), a conserved, PM-localized transmembrane protein (Figure 3). MPRF-induced aminoacylation is under the control of the GraRS two-component regulatory system that increases its resistance against CAMPs. Vancomycin resistance in MRSA is significantly increased due to increased synthesis of lysyl-PG in plasma membrane [70]. Overexpression of MprF in *S. aureus* led to a 35% increase in Lys-PG synthesis, which enhanced the cationic charge of plasma membrane as indicated by the 25% reduction in cytochrome C-binding [71]. MPRF-induced aminoacylation of membrane PLs is observed in many pathogenic bacteria, such as *Bacillus anthracis*, *Bacillus subtilis*, *Mycobacterium tuberculosis*, *Pseudomonas aeruginosa,* and *Rhizobium tropici,* resulting in their multi-drug resistance [72]. PG and CL are lysinated to acquire broad spectrum resistance against CAMPs. In *S*. *aureus*, Lysyl-PG is synthesized in the inner leaflet of PM by lysyl-PG synthase and flipped to the exoplasmic leaflet by a protein translocator termed as lysyl-PG flippase. Evidence demonstrates that both synthesis and flip of lysyl-PG are performed by distinct domains of MPRF: lysyl-PG synthase and lysyl-PG flippase, respectively, which work in concert. The synthase domain that is localized to the C-terminal transfers an activated amino acid from the aminoacyl-tRNA to a PG located in the inner PM leaflet to synthesize the aminoacyl-PG. Structural elucidation of lysyl-PG synthase domain from *Pseudomonas aeruginosa* and *Bacillus licheniformis* demonstrates that it has a tunnel-like active site that binds to the hydrophilic lysyl-t-RNA and the hydrophobic PG on its cytosolic and membrane sites, respectively. The tunnel serves as the pathway for the transfer of lysine moiety from the lysyl-t-RNA to the phospho-glycerol head group of PG [73]. The nascent aminoacyl-PG in the inner leaflet has propelled to the active site of the flippase domain of MPRF to be flipped to the exoplasmic leaflet, thereby populating the PM surface with cationic Lysyl-PG (Figure 3) [74]. Both synthase and flippase domains of MPRF are essential for CAMP resistance in MRSA, as deletion mutant lacking either domain was sensitive to CAMPs [71]. In *Rhizobium tropici*, the flippase domain catalyzes rapid flipping of the lysyl-PG from the cytotolic leaflet to the exoplasmic leaflet through a hydrophilic tunnel gated by two polar amino acids R^304^ and E^280^ [75]. Conformational alteration in the active site of flippase domain with respect to the synthase domain facilitates the rapid diffusion of the lysyl-PG from the synthase domain to the flippase domain. In *C. perfringens*, lys-PG and Ala-PG are synthesized by distinct MPRF paralogs [76]. However, their mechanisms of action and specificity remain elusive. Investigation suggests that both lys-PG and Ala-PG have overlapping function and can be translocated by the same flippase domain of MPRF, leading to their comparable degree of resistance against daptomycin, nisin, and gallidermin [77]. However, the mechanism by which the head groups of Lys-PG and Ala-PG with very different sizes and charges are translocated by the same MPRF flippase remains to be explored. Additionally, how the cationic Lys-PG and neutral Ala-PG exhibit comparable resistance against the CAMPs remains to be solved [74].

MPRF extracts energy from the concentration gradient of Lys-PG, which is higher in the cytosolic site compared to the exoplasmic site, although a scramblase-like activity similar to TMEM16F is also proposed. However, an acidic pH that protonates E^280^ facilitates Lysyl-PG translocation by releasing the energy of protonation that lowers the activation energy. Monoclonal antibody M-C7.1 developed against MPRF that binds to a specific loop in the flippase domain rendered MRSA susceptible to daptomycin. It also impaired MRSA survival in human phagocytes. Hence, MPRF is a novel target for drug-resistant bacteria and its inhibitors are recommended as a novel group of anti-virulence molecules [78].

## 4. Role of Membrane Microdomains in MDR

Lateral heterogeneities in bacterial plasma membrane, termed as microdomains, play a key role in membrane-mediated stress regulation, including antibiotic resistance. Increasing evidence reveals that these membrane microdomains are signaling nano-platforms, the composition, lifetime, size, and dynamics of which regulate bacterial response to antibiotics [79,80,81]. The anionic lipid CL has two small glycerol heads and four fatty acyl tails, giving it a conical topology. Additionally, many non-lipid components such as carotenoids, hopanoids, and exogenously added polycationic compounds stimulate phase separation between anionic and zwitterionic lipids in PM to form membrane microdomains. CL is synthesized in the inner leaflet of cytosolic membrane and equilibrated between both the leaflets through the action of biogenic membrane flippase [31]. However, the PbgA complex, an inner-membrane protein containing five transmembrane helices and a globular periplasmic domain (act as lid) binds CL to promote its PhoPQ-regulated trafficking from the IM to the OM [82]. However, it remains unknown if the same protein is involved in the retrograde transport of CL or if it translocates other PLs, such as PE or PG (Appendix A).

### 4.1. Structural Organization of Membrane Microdomains in Drug Resistant Bacteria

The cone-shaped CL favors the negative curvature or non-bilayer structures of plasma membrane. Hence, CL is often found clustered at polar and septal regions of rod-shaped bacilli, such as *Bacillus subtilis* and *E*. *coli,* which either exhibit a high degree of curvature or serve as the starting point of membrane synthesis [69,83]. Additionally, membrane protein complexes such as components of cell division machinery and electron transport chain, are tightly bound to CL. CL clusters at the polar or septal region of *Enterococci* act as the binding sites for CAMPs, including daptomycin. Although daptomycin is highly effective in the clinic, several non-susceptible *S*. *aureus* and *Enterococci* genetic variants exhibit mutation in genes involved in maintenance of either the cell wall or plasma membrane. Most mutants are clustered in genes involved in teichoic acid alanylation complex DltABCD, the PG synthase (PgsA), the lysyl-PG synthase (MprF), cardiolipin synthase (Cls), cell envelope stress-responsive two-component system (LiaRS), and the cell wall-related two-component system WalKR (YycFG) [84]. Reorganization of CL that is mediated by the gene LiaR depletes the CL targets from septal and polar region, which serves as the principal target sites for daptomycin. Deletion of LiaR abrogates the septal clustering of CL-enhancing dap sensitivity of *Enterococci* [85]. In the case of daptomycin-resistant *Enterococcus faecalis*, single amino acid mutation due to LiaF genes have been discovered in cardiolipin synthase. This particular gene has been proven to encode a transmembrane protein which is a part of a three-component system involved in the regulation of the bacterial cell envelope homeostasis in response to stress. As mutations occur within this region, daptomycin no longer interacts with the septum and subsequently gets diverted and trapped in distinct membrane domains. Daptomycin demonstrated increased binding to the bacterial membrane in presence of CL [86]. Daptomycin induces lipid reorganization in bacterial plasma membrane, leading to inhibition of lipid II synthase (MurG) and phospholipid synthase PlsX. Those are predominantly localized to fluid membrane microdomains. As daptomycin reorganizes, the fluid domains into more hydrophobic liquid crystalline domains, and these proteins get structurally and functionally altered, leading to inhibition of cell wall synthesis and membrane biogenesis. Further, mismatch between fluid and more hydrophobic rigid membrane domains results in proton leakage and rapid cell lysis [84]. As increased CL content decreases the effective concentration of PG and increases membrane rigidity, it inhibits transbilayer flipping of daptomycin, preventing its membrane insertion [87]. Conformational flexibility of the glycerol head groups in CL is very restricted compared to other PLs. Hence, increase in CL content reduces the membrane binding of the above peptides, leading to significant reduction in their membranolytic activity. In *E*. *coli*, a threefold increase in CL synthesis during the stationary phase enhances their resistance against vancomycin. This CL-induced increased drug resistance is primarily due to the enhanced activity of the multi-drug resistance transporter MsbA, an ATP-dependent flippase [88]. Additionally, CL regulates the transport of LPS and lipid A to the outer membrane in gram-negative bacteria; those are the key constituents for biogenesis and maintenance of OM.

Although cholesterol is absent from bacterial plasma membrane, presence of its structural analogs such as cholesterylglycosyl esters and squalene like molecules in many pathogenic bacteria create raft-like domains. *B*. *subtilis* contains two flotillin-like proteins, floT and floA, which form dynamic membrane microdomains. Microdomains in *B*. *subtilis* regulate morphogenetic processes such as sporulation, biofilm formation, and cell division [89]. Plasma membrane of the tick-borne spirochete *Borrelia burgdorferi*, which is responsible for causing Lyme disease, contains phosphatidycholine, phosphatidylglycerol, and lipoproteins. Additionally, it has free cholesterol and two cholesterol glycolipids: (i) acylated cholesterylgalactoside (ACGal) and (ii) cholesterylgalactoside (CGal), in addition to a non-cholesterol glycolipid, monogalactosyldiacylglycerol (MGalD) [90]. *Borrelia burgdorferi*, although it is devoid of its own sterol biosynthetic machinery, derives these sterols through direct contact with the host plasma membrane and host-derived vesicles. Membrane microdomains of *Borrelia burgdorferi* also contain raft-associated proteins OspA and OspB, which are membrane domain stabilizers [91]. Staphyloxanthin, the golden carotenoid pigment in plasma membrane of *S*. *aureus* induces phase separation and forms microdomains. STX pigments exhibit high affinity to CL and flotillin-like proteins to form raft-like domains in *S*. *aureus*. Flotillins such as floA and floT were observed in the septum of dividing *B*. *subtilis,* indicating that they interact with septum-localized proteins. Laser-based photo disassembly of staphyloxanthin membrane microdomains revives the conventional antibiotics against MRSA [92].

### 4.2. Membrane Microdomains as Novel Antibiotic Targets against MDR Strains

CL is a membrane stabilizer under adverse environmental conditions. Increasing evidence suggest that increase in CL content and its reorganization in bacterial plasma membrane are acutely regulated in response to antibiotics [93]. For instance, Aurein 1.2, a short peptide that has the tendency to induce positive curvature in target bacterial membrane has been proven to induce membrane destabilization and lysis [94]. However, increase in CL content counteracts the membrane-destabilizing effect of aurein 1.2 by inducing negative curvature [95]. Structure and surface charge density of CL, plays a key role in reducing the susceptibility of the POPG/TOCL(2-oleoyl-1-palmitoyl-sn-glycero-3-phospho-rac-(1-glycerol)/(1,1′,2,2′Tetraoleoylcardiolipin) bilayer to peptide aurein 1.2. The lytic activity of many antimicrobial peptides, such as magainin 2, polybia-MP1, LL-37, and ΔM2, is impaired by an increase in CL content [95]. Cyclo (RRRWFW), a new synthetic antibiotic, exhibits a unique mechanism of action by binding to plasma membrane that stimulates redistribution of membrane lipids. Peptide-induced lipid redistribution results in formation of membrane domains, leading to altered membrane biophysical properties and cell lysis [96]. Telomycin, a cyclic depsipeptide, and its analog LL-A-0341β, produced by *Streptomyces canus,* are presumed to target CLs in the plasma membrane of Gram-positive bacteria, resulting in membrane destabilization and cell lysis [97]. In *P. aeruginosa* plasma membrane, CL organizes into clusters that maintain the structural integrity of IM proteins including cytochrome c oxidase and succinate-ubiquinone oxidoreductase [98]. The cationic aminoglycoside 3′,6-dinonyl neamine interacts with *P*. *aeruginosa* plasma membrane in a CL-dependent manner to increase its permeability and rigidity. Additionally, it inhibits the respiratory chain complex, alters shape of the bacterium and increases membrane curvature through its interaction with MreB, a protein involved in maintenance of cellular shape [98]. 3′,6-dinonyl neamine-induced the dehydration of membrane bilayer and pore formation inLUV models that correlated with the molar concentration of CL [99]. An *E. coli* strain lacking CL synthase A is almost completely resistant to sphingosine, an amino alcohol that induces rapid lysis of bacterial plasma membrane compared to the wild type. Sphingosine induces rapid lysis of *P. aeruginosa* and *S. aureus* due to a massive increase in membrane permeability. Findings suggest that CL is the primary target of sphingosine that induces clustering of CLs at a lower concentration. However, absence of CL in the membrane requires elevated concentration of sphingosine to cluster smaller negatively charged lipids into larger rigid domains [100].

Staphyloxanthin domains in MRSA are the platforms that facilitate protein oligomerization and interaction, including the penicillin-binding protein PBP2a, to further promote virulence and antibiotic resistance [92]. However, photo-disassembly of staphyloxanthin domains increases the bacterial sensitivity to antibiotics, opening new avenues for combinatorial phototherapy against MRSA. CL inhibits the pore-forming activity of daptomycin in a liposome model, indicating that bacteria may become more resistant to daptomycin by increasing the CL content in their cell membranes. Inclusion of 10% CL in model membranes was sufficient to effectively suppress membrane translocation and pore formation in liposomes. In presence of 10% CL daptomycin insertion into PC/PG lipid monolayers is deeper and/or more stable as indicated by an increase in surface pressure. The surface pressure differential decreases again when CL concentration is increased to 20%. CL restricts daptomycin to the OM leaflet that prevents it from reaching out to the inner membrane leaflet, thereby decreasing pore formation [87].

## 5. Membranotropic Phytochemicals as Potential Drug Leads against Bacterial MDR 

Phytochemicals are a diverse group of naturally occurring compounds that are currently being repurposed as potent drug leads. Amphiphilic phytochemicals with membrantropic activities interact with bacterial plasma membrane to alter its physicochemical properties, such as fluidity and permeability. However, many phytochemicals exhibit their antibacterial activity through multiple mechanisms.

### 5.1. Phytochemicals That Increase Plasma Membrane Permeability [101,102,103,104,105,106,107,108,109,110,111,112,113,114,115,116,117,118,119,120,121,122,123,124,125,126,127,128,129,130,131,132,133,134,135,136,137,138]

Most amphiphilic phytochemicals such as terpenes, flavones, and polyphenols exhibit antimicrobial activity through interaction with plasma membrane of the target microbe (Table 1). These phytochemicals interact with PM lipid bilayers to alter packing of membrane lipids, organization, and fluidity. Antimicrobial activity of a phytochemical is dependent upon its molecular weight, chain length and lipophilicity. Lipophilicity which that is defined as Log (K_o/w_), where K = the partition coefficient between octanol (O) and water (w), determines its capacity to integrate with bacterial plasma membrane due to high steric effect of the compound [101]. Presence of alcoholic functional groups and double bonds favor membrane association and disruption [102,103]. Synthetic flavonoid derivatives such as 4-chromanone exhibited significant antimicrobial activity against MRSA in presence of phenol –OH groups at 5′ and 7′-position [104]. Similarly, O-prenylation for 4- and 3-isomers of coumaric acid increases its anti-tuberculosis potential by increasing its membrane permeability [105].

### 5.2. Phytochemicals That Alter Membrane Fluidity

The increased fluidity changes the topology of membrane proteins and induces disturbance in the respiration chain [16]. Lipid oxidation alters membrane fluidity, leading to its disruption and massive leakage of cell contents [106]. Studies revealed that the presence of a lipophilic group at position 8 improves the antibacterial activity of flavonoids by enhancing membrane fluidity [107] (Table 2).

**Table 1 membranes-12-00914-t001:** Membrantropic phytochemicals that exhibit antibacterial activity through alteration of plasma membrane properties.

Phytochemical	Source Plant	Bacteria (MIC/MBC)	Mechanism of Action	Reference
Quinoa Saponins	*Chenopodium quinoa*	*F*. *nucleatum* (MIC = 31.3 μg/mL & MBC = 125 μg/mL)	Disruption of plasma membrane.	[116,117]
Sophoraflavanone G & B	*Sophora exigua*, *Sophora flavescens*	*S. aureus* (MRSA) (MIC = 15.6 to 31.25 μg/mL)	Membranotropic and lipophilic.	[118,119]
Thymol and Gallic acid	*Punica granatum*, *Camellia sinensis*	*E. coli* (600 μg/mL), *P. aeruginosa* (500 μg/mL)	Adjuvant with antibiotics leads to LPS disintegration.	[120]
Kamferol and Quercetin	*Persea lingue*	MRSA (MIC = 128–256 μg/mL)	Increased permeability.	[107,109]
Emodin	* Rhamnus * spp.	*E. coli* (MIC = 2.2 µM)	Formation of non-bilayer phases resulted in membrane leakiness.	[124]
Barbaloin	* Aloe vera *	*E. coli* (MIC = 2.8 µM)	Induction of membrane leakiness through promotion of gel-fluid phases.	[124]
Terpenes (α-terpineol, linalool)	*Mentha Spicata* and *Lavendula augustifollia*	*E. coli *and* P. aeruginosa* (MIC = 2000 µg/mL)	Membrane disruption.	[102,103]
4-chromanone	*Lasiolaena morii*	*E. faecalis*, *S. aureus*, *M. tuberculosis* and *C. Difficile* (MIC = 3.13–6.25 μg/mL)	Decrease membrane potential.	[104]
Chalcones	*Lophira alata*	*E. faecalis*, *S. aureus*, *M. tuberculosis* and *C. Difficile* (MIC = 1.56–3.13 μg/mL)	Decrease membrane potential.	[104]
Olympcin A	*Hypericum olympicum*	*E. faecalis*, *S. aureus*, *M. tuberculosis* and *C. Difficile* (MIC = 1−2 μg/mL)	Decrease membrane potential.	[104]
Cinnamic acid (4-Coumaric acid)	*Liquidambar orientalis*	*M. tuberculosis * (MIC = 844 μM)	Increase membrane permeability.	[105]
Chanoclavine	*Ipomoea muricata*	*E. coli* (MIC = 125 μg/mL)	Down-regulates expression of efflux pumps and up-regulation of porin, increasing membrane permeability.	[134]
Berberine	*Berberis vulgaris*	*S*. *aureus* (≥128 μg/mL)	Increased the permeability of cell membrane and deteriorated the integrity.	[135,136]
3-p-Trans-coumaroyl-2-hydroxyquinicacid	*Cedrus deodara*	*S. aureus* (2500–10,000 μg/mL)	Conformational changes in membrane protein.	[122]
p-Coumaric acid	*Arachis Hypogaea* and *Solanum lycopersicum*	*S. dysenterae* (10 μg/mL), *E. coli* (80 μg/mL) *S. typhimurium* (20 μg/mL)	Increase in permeability of bacterial cell membranes and K^+^ ion release.	[127,128]
Curcumin I	*Curcuma longa*	*S. aureus* (200 μM), *E. coli* (100 μM)	Increased membrane leakiness.	[125,126]
Epicatechins	* Camellia sinensis *	-	Disruption of membrane lipid bilayer.	[113]
EGCG	* Camellia sinensis *	*P. aeruginosa* (0.2–0.4 mg/mL) *S. mutans* (MIC = 0.125 mg/mL)	Solubilize lipid molecules from the bilayer, resulting in decreased lipid packing.	[111]
Farnesol	Symbopogon and Citronella	*S. aureus* (MBC = 40 μg/mL)	Increased initial and total leakage of K^+^ ions.	[137]
Nerolidol	Symbopogon	*S. aureus* (MIC = 512–1024 μg/mL, MBC = 80 μg/mL)	Leakage of K^+^ ions	[137]
Thymol	*Thymus vulgaris*	*S. saintpaul* (MIC = 49.37 μg/mL) *P. aeruginosa* (MIC = 5–8 µg/mL)	Amine and hydroxylamine groups of the proteins on bacterial membrane altering their permeability.	[129,130]
Carvacrol	*Thymus capitatus*	*E. coli* (MIC = 8 μg/mL), *E. aerogenes* (MIC = 8 μg/mL) *S. aureus* (MIC = 7 μg/mL) *P. aeruginosa* (MIC = 7 μg/mL)	Increasing membrane permeability.	[130,131]
Eugenol	*Syzygium aromaticum* and Cinnamon	*H. pylori* (MIC = 2 μg/mL), *S. typhimurium* [0.0125% (*v*/*v*)]	Membrane expansion, increased membrane fluidity and permeability.	[132,133]
Cinnamaldehyde	* Cinnamomum* *ceylanicum*	*S. aureus* (MIC = 2 μg/mL), *H. pylori* (MIC = 15 μM), *E. coli* (MIC = 7.6 μM), *P. aeruginosa* (MIC = 10.6 μM)	Disruption of membrane integrity by increasing permeability.	[138]
Gymnemic acid	*Gymnema* *sylvestre*	*P. aeruginosa* (IC_50_ ˂ 100 µg/mL) *S. aureus* (IC_50_ ˂ 350 µg/mL) *E. coli* (IC_50_ = 500 µg/mL)	Flip-flop of a fluorescent-labeled phospholipid analog NBD-phosphatidyl ethanolamine (NBD-PE) in the GUVs.	[115]

‘-’, not available, Abbreviations: POPE, Palmitoyloleoylphosphatidylethanolamine; POPG, Palmitoyloleoylphosphatidylglycerol; POPC, Palmitoyloleoylphosphatidyl choline; DPPC, Dipalmitoylphosphatidylcholine and DOPC, Dioleoylphosphatidylcholine; GUV, Giant Unilamellar Vesicles; Egg-PC, Egg Phosphatidylcholine; MIC, Minimum Inhibitory Concentration; MBC, Minimum Bactericidal Concentration; CH, Cholesterol; SOPS, Stearoyl-oleoyl-phosphatidylserine.

**Table 2 membranes-12-00914-t002:** Phytochemicals that act through alteration of membrane fluidity.

Phytochemical	Source Plant	Model Membrane/Bacteria	Measurable Parameter	Effects on Membrane	Ref.
Sophoraflavanone G and Naringenin	*Sophora exigua*	DPPC and POPC liposomes	Fluorescence polarization	Decreases fluidity due to presence of lavandulyl group at the 8-position and 5-, 7- and 4′-hydroxylation. Increase in polarization with ANS and PNA.	[121]
Linalool; 1,8-Cineol; α-Terpineol	*Coriandrum sativum*	*S. aureus* and *E. coli* cells	Scanning electron microscopy	Increase the fluidity and permeability.	[103,121]
Gallic acid, Methyl Gallate and Alkyl gallate	*Bahunia kockiana*	MRSA cells (250–500 μg/mL)	Scanning electron microscopy	Decrease in fluidity alter the membrane permeability	[106]
3-p-trans-Coumaroyl-2-hydroxyquinic Acid	*Cedrus deodara (pine needles)*	*S. aureus* cells	Membrane potential measurementand Flow cytometry	Increase membrane fluidity due to decrease in fluorescence polarization of DPH. Disruption of cell membrane led to leakage of intracellular constituents.	[122]
Epicatechin; -Epigallocatechin;	*Camellia sinensis*	DPPC:DOPC liposome	Fluorescence polarization	Decrease in fluidity results in antiplaque and hepatoprotective effects of green tea.	[123]
Proanthocyanidins	*Vaccinium macrocarpon* and *Vitis vinifera*	DPPC; DOPC liposomes, POPC: POPE: SPM: CHOL =1:1:1:2 (60 μg/mL), *S. aureus* cell	Fluorescence polarization	Disrupt the membrane integrity by increasing cell membrane fluidity. Decrease in FP.	[139]
Ajoene	*Allium sativum*	Phospholipid/cholesterol unilamellar vesicles (2.5 μM)	Electron spin resonance	Increase the fluidity of the hydrocarbon chains. Increase in DPH polarization.	[16]
Capsaicin; *N*-Vanillylnonanamide	*Capsicum* spp.	Bacterial cell mimetic membranes (100–500 μM)	Fluorescence polarization	Decrease in fluidity due to decrease in PNA polarization.	[140]
Baicalein	*Scutellaria baicalensis*	*E. coli* cell (70.94 μg/mL)	Fluorescence polarization	Decrease in membrane fluidity by reducing membrane polarity.	[107]
Cinnamaldehyde	Cinnamon	*S. putrifaciens* cell (414 μg/mL)	Fluorescence polarization	Alteration in membrane structure due to increased membrane fluidity and decreased membrane polarity.	[141]
Tangeritine	*Citrus sinensis*	LUV of DPPC and DPPG, *E. coli* cell	Fluorescence polarization	Methoxyl group at C-8 in the A ring which makes it more lipophilic and decrease membrane polarity, thereby increase membrane fluidity.	[107]

‘-’, not available. **Abb.**: SPM Specialized pro-resolving lipid mediators; CHOL, Cholesterol; MRSA, Methicillin Resistant *S. aureus*; LUV, Large Unilamellar Vesicles.

### 5.3. Modulators of Membrane Protein Activity

Membrane proteins such as OM porins and efflux pumps are modified by several phytochemicals. These phytochemical modulators result in compromised integrity, increased permeability, changes in confirmation, increased leakage of cytosolic contents including ions, metabolites, and proteins, inhibits H^+^-ATPase in cytoplasmic membrane, and distorts morphology in MDR pathogens [108].

### 5.4. Phytochemicals That Alter Membrane Domain Organization

The membrane interaction of flavonoids is influenced by the number and the position of –OH groups. 3′–OH on the C ring is important for increased membrane interaction of flavonoids. The stringency of membrane interaction of flavonoids in the order kaempferol > chrysin and quercetin > luteolin corroborates a previous study that C_3_–OH is the primary determinant for significant membrane interaction. Polymethoxy flavones increase membrane fluidity compared to its parent flavonoid molecule, resulting in the formation of membrane microdomains. It is likely that the observed differences might be a result of increased binding affinity of polymethoxyflavones to the liposomes compared to flavonoids resulting in the formation of membrane microdomains [109]. Membranotropic activities of catechins are mostly due to their ability to penetrate membranes and form hydrogen bonds with lipid head groups, resulting in intercalation between lipids and bilayer expansion [110]. EGCG at 30 µM induces leakage in GUVs by solubilizing membrane [111]. Membrane interaction of catechins is dependent on the number of –OH groups on B ring and its stereochemical configuration, cis-types being better membranotropic agents than the trans-types. Increased negative membrane charge reduced membrane interaction of catechins [112]. Alkyl-catechins with 5–7 carbon fatty acyl chains exhibited increased membranolytic activity compared to those with longer fatty alkyl chains [113]. Accordingly, (-) epicatechingallate and (-) epigallocatechingallate exhibited ~1000 times more affinity for 1,2-dimyristoyl-sn-glycero-3-phosphocholine bilayer compared to (-) epicatechin and (-) epigallocatechin [114]. These findings reveal that amphiphilic phytochemicals exhibit greater membranotropic activity compared to their hydrophilic and hydrophobic counterparts. Phytochemial-induced flip-flop of membrane lipids is one of the primary mechanisms of action that results in increased membrane permeability [115]. Further, promiscuous alteration of membrane protein activities driven by membrane association of phytochemicals is a primary mechanism of their antibacterial action [24].

## 6. Conclusions and Future Prospective

Despite the cutting-edge advancements made in the healthcare sector, discovery of antibacterial drugs is a never-ending global health issue due to the proficient bacterial adaptation to new antibiotics. Membrane-targeted compounds are amongst the most advanced group of antibiotics against bacterial pathogens. However, the selection pressure imposed by indiscriminate use of antibiotics leads to genetic variation that spreads rapidly within bacterial population through horizontal gene transfer mechanisms. Lipid components such as LPS and plasma membrane microdomains are being explored as novel drug targets. Additionally, the protein components such as Mla complex, MPRF, Lpts, lipid II flippase, PL synthases, and flippases, in addition to their corresponding genes, are the most sought after drug targets for development of current generation drugs. However, most of their structural details and mechanism of action remain elusive at present. Several questions pertaining to the role of proteins involved in bacterial membrane biogenesis demand the need for further investigation. For example, given the role of Mla complex in maintenance of OM asymmetry, it remains unclear how the PLs are translocated between MlaA, MlaC, and MlaBDEF complexes. We have comprehensively discussed the physiological importance of PL translocation in bacteria cell envelopes. In this context, it is important to understand the factors that regulate the switching of PL translocation from anterograde to retrograde upon their mislocalization in OM outer leaflet. What also remains unclear is whether the same Mla complex translocates all PLs or exhibits specificity towards a single PL. We do not know how the ala-PG and Lys-PG with different sizes and charges induce comparable resistance against CAMPs. It is also important to identify whether there are any additional and yet unexplored resistance factors that work in concert with MPRF. Hence, exploration of the membrane lipidome, proteome, and their interactions/functional cross-talks in drug-resistant microbes constitutes the basis for development of new generation antibiotics and novel targeting strategies. Given the global urgency, there is a need for innovation in research and development to overcome and mitigate the systemic impact of antimicrobial drug resistance. Thus, exploration of membranotropic phytochemicals and their synthetic derivatives as herbal alternatives to target bacterial membrane components may pave the way for innovative phytotherapeutics to combat bacterial multi-drug resistance. Enhanced membrane permeability of the therapeutic compounds constitutes a key step towards the successful drug development. Synthetic conjugates of membranotropic phyto-compounds with antimicrobials having intracellular target will enhance their membrane permeability and therapeutic efficacy. However, investigation on synthesis of phytoconjugates and their clinical trials is limited due to the lack of high throughput purification and identification of membranotropic phytochemicals. In addition, the interaction of membranotropic phytochemicals with bacterial membranes is species-specific and depends upon the membrane lipid composition. Our comprehensive review reveals that repurposing of membranotropic phytochemicals for enhanced targeting of therapeutics will unveil novel sustainable herbal therapeutic strategies against overcoming the impact of drug-resistant bacteria.

## Figures and Tables

**Figure 1 membranes-12-00914-f001:**
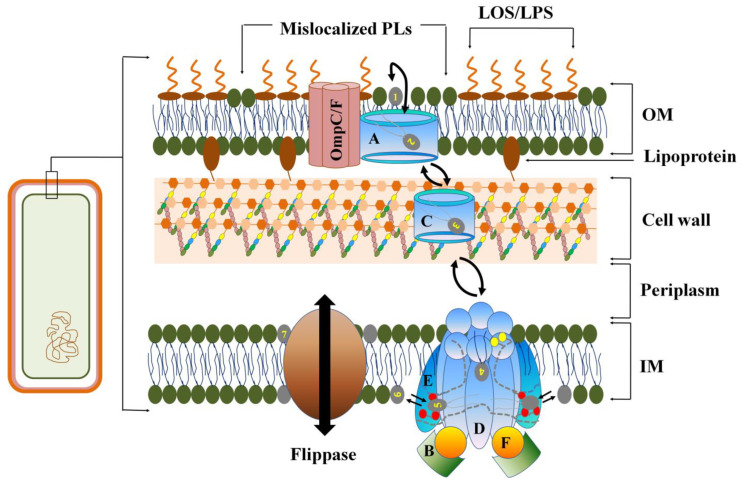
Role of maintenance of lipid asymmetry (Mla) complex in asymmetric distribution of LPS and PLs in the outer (exoplasmic) and inner (periplasmic) leaflets of Gram-negative OM. In Gram-negative bacteria, the inner membrane (IM) is a symmetric bilayer of phospholipids (PLs), whereas the OM is asymmetric with LPS/LOS localized exclusively to the outer leaflet and PLs localized to the inner leaflet. The Mla complex in *Acenetobacter baumanii* is a six-component protein complex that forms three distinct units: MlaA (A), MlaC (C), and MlaBDEF (B, D, E and F), out of which, MlaA is localized to OM, MlaC is localized to the periplasm, and MlaBDEF complex is localized to the IM. The unique structural features of MlaA enables it to act like a vacuum cleaner to remove any mislocalized PLs from the OM outer leaflet to MlaC that shuttles the mislocalized PLs into the PL binding site of MlaBDEF complex that is oriented toward the periplasm. The route of PLs through IM is formed by the transmembrane domains of MlaD and MlaE as depicted by dotted lines. The selectivity of PLs through the MlaBDEF complex is determined by Leu153/Leu154 (yellow balls) at the periplasmic end and R47, R14, and R234 (red balls) near the cytosolic end. ATP binding sites are present on the MlaF subunits near the MlaE-F interface. The PL pathway through MlaBDEF complex is bilaterally symmetrical, making the PLs use both the routes indiscriminately. From the cytosolic site of the MlaBDEF, the PLs diffuse into the cytosolic leaflet of IM which are equilibrated between both the leaflets facilitated by the biogenic membrane flippases.

**Figure 2 membranes-12-00914-f002:**
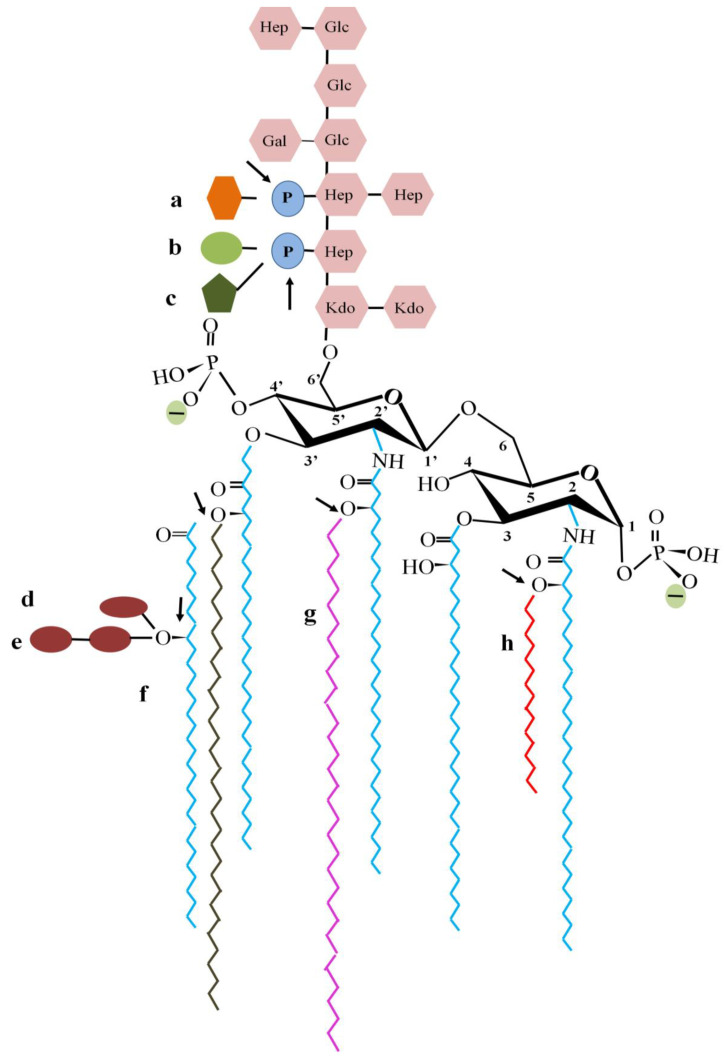
Lipid A modification in Gram-negative bacteria leading to altered membrane properties. Gram-negative bacteria modify phosphate groups with (**a**) phosphoethanolamine, (**b**) glucosamine, and (**c**) aminoarabinose. The fatty acyl tails add (**d**) positively charged glycine, (**e**) diglycine moieties, or (**f**) palmitate at the 3′-position of the glucosamine disaccharide. Fatty acyl chains such as laureate (**g**) and hydroxy–myristate (**h**) are added at positions 2 and 2′, respectively. Arrows indicate the position of the groups modified.

**Figure 3 membranes-12-00914-f003:**
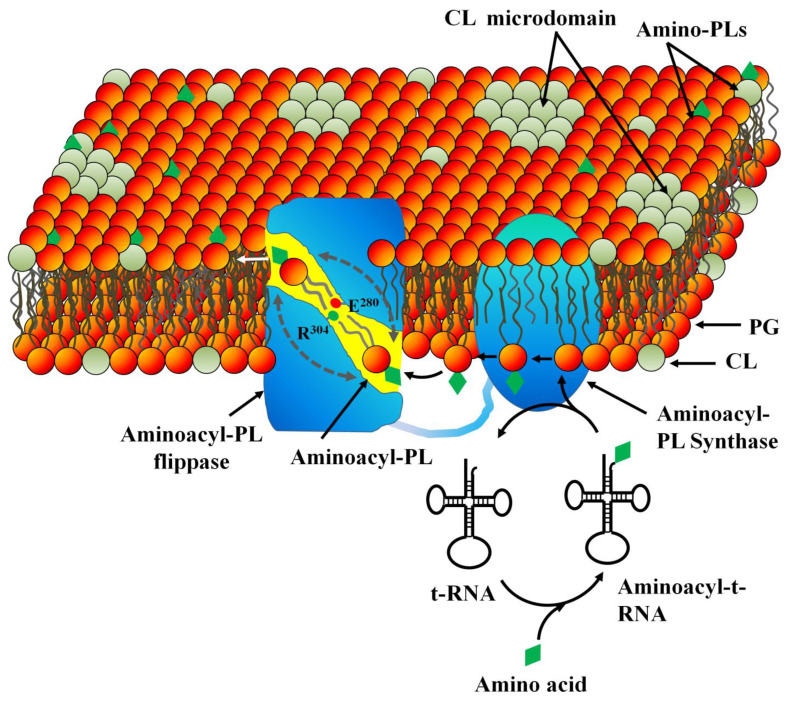
Modification of bacterial inner-membrane lipids leading to multi-drug resistance. The negatively charged PLs in IM of MDR Gram-negative bacteria (e.g., *P*. *aeruginosa*) and the only membrane of Gram-positive bacteria (e.g., *S*. *aureus*) are modified by addition of aminoacyl groups such as alanine and Lysine. The multiple peptide resistance factor (MPRF), a protein localized to the PM, possesses two distinct domains: aminoacyl-PL synthase and aminoacyl-PL-flippase, which work in concert. The aminoacyl-PL synthase domain catalyzes the transfer of an activated aminoacid (green rhombus) such as Ala or Lys to a PG or CL bound to the active site of MPRF near to the inner leaflet of IM. The aminoacyl-PL diffuses to the active site of the flippase domain; from there it is flipped to the outer leaflet through a lipophilic pathway that is gated by two charge amino acids: E280 and R304. The gating mechanism greatly decreases the free energy of translocation plausibly by replacing the water of hydration. The lipid molecule is inverted 180° (indicated by double-headed arrow) within the MPRF to diffuse into the outer leaflet, populating it with aminoacyl-PLs. Increased aminoacyl-PL content in the outer leaflet alters the membrane physical properties such as charge and fluidity, decreasing its affinity for the CAMPs such as daptomycin or other membranolytic antimicrobials. In many bacteria, clustering of CLs into lateral heterogeneities termed as microdomains makes the membrane rigid and less sensitive to membrane-targeted antibiotics.

## Data Availability

Not applicable.

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
