# Peer review of "Harnessing the Role of Bacterial Plasma Membrane Modifications for the Development of Sustainable Membranotropic Phytotherapeutics"

_membranes, 2022, doi:10.3390/membranes12100914_

Round 1
Reviewer 1 Report
The review (membranes-1890375) summarized the recent development of membranotropic phytochemicals and their derivatives targeting bacterial plasma membrane modifications. This is an interesting review topic. However, the authors detail the role of lipopolysaccharides and phospholipid in bacterial multidrug resistance. The summary of new and effective plasma membrane-based target design is insufficient. The figures showed in the paper also needs major adjustments. Many writing mistakes also need be revised in this version.
1. Line 136: The order of the article needs to be adjusted.
2. Line 137-139: The description of the bacteria should be italic.
3. Line 168-169: The abbreviation of LPS has been mentioned before and needs to be deleted.
4. Line 169-170: “Mla (maintenance of outer membrane lipid asymmetry) system” should be changed as “maintenance of outer membrane lipid asymmetry (Mla) system”.
Author Response
Reviewer 1
- Comment
The review (membranes-1890375) summarized the recent development of membranotropic phytochemicals and their derivatives targeting bacterial plasma membrane modifications. This is an interesting review topic. However, the authors detail the role of lipopolysaccharides and phospholipid in bacterial multidrug resistance. The summary of new and effective plasma membrane-based target design is insufficient. The figures showed in the paper also needs major adjustments. Many writing mistakes also need be revised in this version.
Response
The present article has two parts. Part 1 is aimed at the role of different modifications of bacterial membrane components in resisting anti-bacterial drugs and the part 2 is aimed at using plant-derived natural molecules that target bacterial plasma membrane. However, although large amounts of literature describing antibacterial effects of phytochemicals exist, the phytochemicals targeting bacterial plasma membrane that alters a defined component/property of PM is poorly investigated. Although phytotherapeuics are used as herbal drug-leads and widely used across the globe, design of specific membrane-targeted phyto-drugs is limited. Many literatures describe the synthesis of green nanoparticles using solvent extracts, information on synthesis of nanoparticles using single purified phytochemicals is poorly. This review has summerized the phytochemicals that alter membrane properties such as fluidity and permeability which could increase the targeting efficacy and bioavailability of antibacerials. However, our review provides a list of phytochemicals that will enable synthesis or modification of these compounds for future herbal drugs at industrial scale. Figures were rechecked and modified. The manuscript was checked for English and typographical errors
Figure
- Line 137: The order of the article needs to be adjusted.
Response: The order of the article is verified and adjusted
- Line 139-140: The description of the bacteria should be italic.
Response: Bacterial names have been italicized.
- Line 168-169: The abbreviation of LPS has been mentioned before and needs to be deleted.
Response: Lipopolysaccharide is deleted.
- Line 170-171: “Mla (maintenance of outer membrane lipid asymmetry) system” should be changed as “maintenance of outer membrane lipid asymmetry (Mla) system”.
Response: The change was done as directed
Reviewer 2 Report
In this study, the authors outline the role of bacterial membrane lipidome and proteome as well as their organization on multidrug resistance and propose that membranotropic phytochemicals are potential drug leads against MDR bacteria. Overall, this study provides some insightful perspectives, but there are some significant issues that need to be addressed.
Major comments:
1. Does the Mla complex consist of a six-component protein complex in all Gram-negative bacteria?
2. How Gram-negative bacteria identify and locate mislocalized PLs?
3. line 313-315: mcr1 is phosphoethanolamine transferase gene, what is the relationship between mcr1 and L-Ara4N?
4. Line 567: the authors mentioned that antimicrobial activity of phytochemical is dependent upon its size and lipophilicity. Thus, how its size affects the antimicrobial activity of phytochemicals?
Minor comments:
1. Please check and modify the spacing between words in the full text.
2. Please check and correct any misrepresentations in the manuscript. For example, lines 202, please delete “resulting in”; Line 345, please delete “and”; Line 347, please check the spelling of “PhoQ”;
3. Line 30: change “Multi Drug Resistance” to “Multidrug Resistance” or “Multi-drug Resistance”
4. Line 81: what dose “PlaA” refer to? please supplement the unabbreviated form of PlaA in “Abbreviations” section.
5. Please specify the location of Leu153/Leu154 (yellow balls) and R47, R14 and R234 (red balls) in Figure 1.
6. The numbering of titles and subheadings in the manuscript is confusing, please correct it.
7. Line 562: “Phytochemicals that reduce…” reduce or increase, please correct it.
Author Response
Comment: Does the Mla complex consist of a six-component protein complex in all Gram-negative bacteria?
Response: The Mla system is conserved in Gram-negative bacteria, the sequence identity between the MlaBDEFab and MlaBDEFec proteins is around 30–40% identical (depending on the protein, Supplementary Fig. 9), and it is therefore possible that the differences observed between the two structures correspond to variability between bacterial species. (Mann et al., 2021). The maintenance of lipid asymmetry (Mla) proteins (Mla A-F) system is conserved in Gram-negative bacteria and plays an important role in phospholipid transport (Chi et al., 2020).
References
- Chi, X., Fan, Q., Zhang, Y., Liang, Ke, Wan, Li, Zhou, Q., Li, Y., Structural mechanism of phospholipids translocation by MlaFEDB complex, Cell Res. 2020, 12, 1127–1135.
- Mann D, Fan J, Somboon K, Farrell DP, Muenks A, Tzokov SB, DiMaio F, Khalid S, Miller SI, Bergeron JR. Structure and lipid dynamics in the maintenance of lipid asymmetry inner membrane complex of A. baumannii. Communications biology. 2021 ,4,1-9.
Comment: How Gram-negative bacteria identify and locate mislocalized PLs?
Response: The unique structural features of MlaA enables it to acts like a vacuum cleaner to remove any mislocalized PLs from the OM outer leaflet to MlaC that shuttles the mislocalized PLs into the PL binding site of MlaBDEF complex that is oriented towards the periplasm (Figure 1) (our manuscript). (Shrivastava and Chng., 2019). Only PLs localized to the leaflet OM will be directed to the PL binding site of MlaA. However, PLs localized to the inner PM leaflet will not be affected because of the structural features of MlaA. The glycolipids in OM outer leaflet will not be removed because of the high selectivity of MlaA for PLs (Chong et al., 2015).
References
- Shrivastava R, Chng SS. Lipid trafficking across the Gram-negative cell envelope. Journal of Biological Chemistry. 2019,294,14175-84.
- Chong ZS, Woo WF, Chng SS. OsmoporinOmpC forms a complex with MlaA to maintain outer membrane lipid asymmetry in Escherichia coli. Molecular microbiology. 2015,98,1133-46.
Comment: Line 313-317: mcr1 is phosphoethanolaminetransferase gene, what is the relationship between mcr1 and L-Ara4N?
Response- In absence of colistin resistance gene mcr1,modification of Lipid A-Ara4N act as an alternate pathway for colistin resistance mediated by ArnA_ DH/ FT, UgdH, ArnC and ArnT genes (Masood et al., 2021)
References
- Masood KI, Umar S, Hasan Z, Farooqi J, Razzak SA, Jabeen N, Rao J, Shakoor S, Hasan R. Lipid A-Ara4N as an alternate pathway for (colistin) resistance in Klebsiella pneumonia isolates in Pakistan. BMC research notes. 2021 14, 1-7.
Comment: Line 575: the authors mentioned that antimicrobial activity of phytochemical is dependent upon its size and lipophilicity. Thus, how its size affects the antimicrobial activity of phytochemicals?
Response: The size and lipophilicity of phytochemicals determine the membrane permeability of the phytchemical. Hence, the size and lipophilicity are the key determinants of the bioavailability and effective intracellular concentration of the phytochemical in bacteria. The molecular weight, chain length and functional groups are the size determinants that determine their membranotropic and membranolytic activity. For example, emodin compound A of plant Vismia laurentii differs from compound C by the length of the aliphatic chain and methoxy substitution. The substitutions at position 2 of emodine derivatives is detrimental to their bactericidal activity while increase in the aliphatic chain length of the methoxy substitution in position 6 is beneficial to the antibacterial activity (Aleves et al., 2004)
Minor comments:
- Please check and modify the spacing between words in the full text.
Response: Changes made as directed
- Please check and correct any misrepresentations in the manuscript. For example, lines 204, please delete “resulting in”; Line 349, please delete “and”; Line 351, please check the spelling of “PhoQ”.
Response: PhoQ has been changed to PhoPQ.
- Line 30: change “Multi Drug Resistance” to “Multidrug Resistance” or “Multi-drug Resistance”
Response: Corrected
- Line 82: what does “PlaA” refer to? please supplement the unabbreviated form of PlaA in “Abbreviations” section.
Response: PlaA has been changed to Mla
- Please specify the location of Leu153/Leu154 (yellow balls) and R47, R14 and R234 (red balls) in Figure 1.
Response: The location of Leu153/Leu154 (yellow balls) and R47, R14 and R234 (red balls) in Figure 1 have been specified
- The numbering of titles and subheadings in the manuscript is confusing, please correct it.
Response: The subheadings were correctly numbered and ordered
- Line 570: “Phytochemicals that reduce…” reduce or increase, please correct it.
Response: ‘Phytochemicals that reduce’ has been changed to ‘Phytochemicals that increase’
Round 2
Reviewer 1 Report
The review (membranes-1890375) summarized the recent development of membranotropic phytochemicals and their derivatives targeting bacterial plasma membrane modifications. This is an interesting review topic. However, the authors detail the role of lipopolysaccharides and phospholipid in bacterial multidrug resistance. The summary of new and effective plasma membrane-based target design is insufficient. The figures showed in the paper also needs major adjustments. Many writing mistakes also need be revised in this version.
1. Line 136: The order of the article needs to be adjusted.
2. Line 137-139: The description of the bacteria should be italic.
3. Line 168-169: The abbreviation of LPS has been mentioned before and needs to be deleted.
4. Line 169-170: “Mla (maintenance of outer membrane lipid asymmetry) system” should be changed as “maintenance of outer membrane lipid asymmetry (Mla) system”.
Reviewer 2 Report
My comments have been addressed.